# Structural Lattice Topology and Material Optimization for Battery Protection in Electric Vehicles Subjected to Ground Impact Using Artificial Neural Networks and Genetic Algorithms

**DOI:** 10.3390/ma14247618

**Published:** 2021-12-10

**Authors:** Alvian Iqbal Hanif Nasrullah, Sigit Puji Santosa, Djarot Widagdo, Faizal Arifurrahman

**Affiliations:** 1Lightweight Structure Laboratory, Institut Teknologi Bandung (ITB), Faculty of Mechanical and Aerospace Engineering, Jalan Ganesha 10, Bandung 40132, Indonesia; sigit.santosa@itb.ac.id (S.P.S.); dwidagdo@ae.itb.ac.id (D.W.); faizal.arifurrahman@gmail.com (F.A.); 2National Center for Sustainable Transportation Technology (NCSTT), Institut Teknologi Bandung, Jalan Ganesha 10, Bandung 40132, Indonesia

**Keywords:** crashworthiness, twisted-octet lattice, topology optimization, genetic algorithm, neural network, battery, electric vehicle, numerical method

## Abstract

A critical external interference that often appears to pose a safety issue in rechargeable energy storage systems (RESS) for electric vehicles (EV) is ground impact due to stone impingement. This study aims to propose the new concept of the sandwich for structural battery protection using a lattice structure configuration for electric vehicle applications. The protective geometry consists of two layers of a twisted-octet lattice structure. The appropriate lattice structure was selected through topology and material optimization using an artificial neural network (ANN), genetic algorithms (GA), and multi-objective optimization with technique for order of preference by similarity to ideal solution (TOPSIS) methods. The optimization variables are the lattice structure relative density, $\bar{\rho }$*,* angle, *θ*, and strength of the materials, $\sigma_{y}$. Numerical simulations were used to model the dynamic impact loading on the structures due to a conical stone mass of 0.77 kg traveling at 162 km/h. The two-layer lattice structure configuration appears to be suitable for the purposes of RESS protection. The optimum configuration for battery protection is a lattice structure with an angle of 66°, relative density of 0.8, and yield strength of 41 MPa. This optimum configuration can satisfy the safety threshold of battery-shortening deformation. Therefore, the proposed lattice structure configuration can potentially be implemented for electric vehicle applications to protect the battery from ground impact.

## 1. Introduction

The development of transportation technology has been characterized by the development of alternative energy to gradually replace fossil fuel owing to issues of non-renewability and the problem of pollutive emissions [1]. One that seems to be ready in the near future is electric vehicles (EV). Nevertheless, there are still many challenges in the technology, especially in the area of batteries and other reserved energy storage systems (RESS) as the main energy source for EV. The main objectives are how to give the battery system of EV a high level of reliability and security.

The primary material of battery cells commonly used in modern electric vehicles is Li-ion. The lightweight and large energy storage density properties are the main reason for the dominance of Li-on batteries in wide-ranging applications. However, Li-on batteries are also well known to be very reactive to other substances. The possible worst-case scenario is a fire or explosion induced by short circuit or leakage caused by damage to a battery’s protective component.

An incident occurred with a Tesla Model S electric car in October 2013 in Washington state. It was reported that the vehicle caught fire after it hit metal debris on a highway [2]. The isolation of each battery module by fire barriers helped to contain and limit the fire’s damage to a small section in the front compartment of the vehicle. Investigation revealed that the fire was caused by the direct impact of a large metallic object in one of the 16 modules within the Model S battery pack.

In the Tesla Model S 2013, the existing six milimeters of ballistic-grade aluminum armor plate protected the battery pack. The existing ballistic shield is spaced at least 10–50 mm away from the bottom surface of battery pack. A lower ballistic shield could be made from other materials such as aluminum alloy, steel, carbon fiber reinforced polymer (CFRP), fiberglass, or polymer. The gap between the ballistic shield and the lower battery pack surface is filled with compressible material to reduce any deformation or damage to the bottom [3]. Responding the incident, Tesla added triple underbody shields that comprise a hollow aluminum bar, a titanium plate, and a solid aluminum extrusion [4]. Following 152 level tests, the shield prevented any damage that could cause a fire or penetrate the existing battery protector [5].

However, six years later, another Tesla car, this time a Model X, caught fire due to a debris hit, which caused damage to the battery [6]. The lesson learned from this incident was that additional protection still cannot entirely stop debris from reaching, hitting, and causing damage to the battery panel. There is always the need to make the bottom battery panel more resilient against impingement incidents.

Previous studies have been conducted on EV battery models to investigate protection of the module. The research to develop a general methodology of predicting a sequence of local indentation followed by piercing and fracture of the bottom structure of a battery pack was done by Xia [7]. The study also evaluated the model further with detailed analysis of a battery cell. Some studies have shown results and conclusions regarding damage to the battery. For axially loaded cells, it was suggested that an 18650 Lithium battery should not be deformed by more than 3 mm to avoid electrical short circuit [8]. Short circuit could also happen by lateral and bending loads due to a crack in the protective shell, which would cause electrolyte spillage that could spark a violent reaction with air [9].

In the development of RESS, several materials and designs have been developed to obtain efficient battery protection structures. A study by Zhu [10] explored the minimizing of battery-shortening as critical damaging effect criteria. Four designs of armor plating for a multi-tunnel, double-layered, NavTruss sandwich and BRAS sandwich configurations were evaluated. The study showed promising results for sandwich configuration which can give 17–57% improvement relative to the baseline model. Furthermore, Halimah et al. [11] designed and analyzed a battery protection system using two types of aluminum sandwich geometry, namely optimized BRAS and NavTruss. Parametric studies were conducted by Nirmala et al. [12] using hybrid fiber metal laminate (FML) as battery protection. Daniel et al. [13] designed protective batteries using plain-weave Carbon Fiber Reinforced Polymer (CFRP).

The better performance of sandwich structures, particularly compared to solid plate layered panels, is very well known and widely applied to obtain efficient lightweight structures [14]. The distinctive structural efficiency of sandwich panel configuration comes mainly from the existence of the core part as the dominant factor for the structural stiffness of the sandwich panel plate configuration. Specifically, for cases against the impact load, the core also plays an important role in absorbing energy through compressive deformation.

In general, cellular materials are suitable for use as material for a sandwich core. Already established cellular materials such as honeycomb or foam materials have shown their applicability to a sandwich core. More recently, lattice material appears to be a new option for cellular material for a sandwich core [15]. The lattice structure architecture concept comes from ideas of using the small-scale truss-like configurations, because the load distribution represents the maximum load path for every elemental lattice member. Thus, naturally, the lattice structure embodies the fundamental principle of structural optimization. However, just after the emergence of recent additive manufacturing technology that enables us to manufacture small-scale lattice configuration, engineering research of lattice structures has picked up pace. The configuration can be used to achieve excellent performance and multi-functionality while reducing component weight. The high strength of lightweight material is key to lattice structures [16]. As a type of cellular structure, the regular periodic arrangement of the lattice structure cells has better structural performance than those with a random void distribution such as foam materials.

The current study presents a new design of lattice structures using a twisted-octet lattice to dissipate collision energy. The choice of twisted-octet type was based on the result of previous optimization studies [17]. The optimization scenario is based on artificial neural networks (ANN) and genetic algorithms (GA) as successfully employed in Pirmohammad et al. [18] to study the crashworthiness of multicell columns. ANN is used to determine the relationship between input (angle, density, and material) and output (SEA and displacement) using sample data that has been obtained from finite-element analysis. Maximizing specific energy absorption and SEA, and minimizing battery-shortening deformation, *δ*, as functions of the geometry and material parameters are the objectives of the current optimization study using the Pareto frontier and TOPSIS method. The sample data were generated using a numerical method with finite-element code LS-DYNA. The numerical models of EV battery protection structure with variation of geometry and material parameters were simulated to receive ground-impact loading. A velocity of 45 m/s applied to the impactor model was defined to include 18% safety factor allowances to the typical 38 m/s velocity of ricocheting debris caused by a car running at 38 m/s speed limit in the US [11]. Therefore, a perpendicular direction of the impactor to the battery bottom panel and a velocity of 45 m/s were defined to represent the worst-case loading.

Using the sample data, the ANN was trained to obtain the weights and biases of approximated functions between the output of SEA and *δ* to the input of geometry and material parameters. The weights and biases were then optimized using GA to reach the smallest NMSE. The validated ANN-GA functions were then used in a decision-making method called TOPSIS (technique for ordering preferences by similarity to ideal solution) to select the best optimal structure from a crashworthiness point of view. The study is limited to a few worst-case scenarios within various possibilities of impact load as affected by impactor velocity, type, and geometry. The thermal expansion effect due to the heat of the battery chemical reaction is not included, in order to simplify assumptions, since the focus of the study is the structural dynamic behavior of the lattice sandwich core.

The study has resulted in not only a new design of lattice materials for battery protection structure, but has also laid an important foundation for further development of battery protection structures using the robust optimization methods of ANN and GA to explore various options of using new materials and topology based on the configuration of cellular materials.

## 2. Design and Finite-Element Modeling

### 2.1. Battery Model

The battery model in this research is a sub-system model of the RESS battery developed by Sahraei et al. [9], Xia et al. [7], Zhu et al. [10], Halimah et al. [11], Nirmala et al. [12], and Irawan et al. [13], based on a commercial EV design. The model consists of a floor panel, battery module, sandwich panel, and impactor. The sandwich panel has an upper layer, lattice structure, and lower layer. The overall model is shown in Figure 1.

The structural component models consisted of floor panels and sandwich plate panels made from aluminum alloy to give a lightweight characteristic to the structures. The components were modeled with shell elements of dimensions 93.3 mm × 93.3 mm. Floor panel thickness was 1 mm, and sandwich panels, which were divided into three parts, had a thickness of 3.175 mm for the upper and lower layers, and 15 mm for the core. The impactor was modeled with a blunt cone-shaped rigid body weighing 0.77 kg, tip radius of 10 mm, and semi-apex angle of 45°. The kinematics impactor was simplified into vertical movements without the influence of rotation.

The battery module model was composed of battery cells and module housing (battery house). A total of nine cells of Li-ion 18650 batteries was arranged 3 × 3 vertically in housing modeled with 3 mm-thick polypropylene plastic, as shown in Figure 2a. The battery model consisted of two parts, i.e., jellyroll and shell casing, which represents the battery skin, as in Figure 2b. The jellyroll represents the inside of the battery, namely the Li-ion, electrode, and separator. The shell casing used a 0.25 mm-thick steel plate. The 18650 battery models have been validated by several studies, including uniaxial compression and lateral loading [8].

The basic geometry and size of lattice structures are shown in Figure 3. The optimized structural design was adapted from Nasrullah et al. [17]. A lattice panel of dimensions 65 mm × 65 mm × 15 mm was arranged as two layers. The component is designed to adsorb as much energy as possible. The geometry of the lattice structure can be seen in Figure 3.

### 2.2. Lattice Design

The earlier optimization study [17] found that the twisted-octet form of lattice configuration had higher specific energy absorption than ten other configurations when subjected to compressive load. Eleven types of lattice configurations were examined to determine the highest specific energy absorption capability, i.e., kagome, tetrahedron, pyramid, cube, truncated pyramid, octahedron, rhombicuboctahedron, rhombic dodecahedron, open-cell, and octet-lattice structures. Numerical analysis and design optimization was performed on a single cell unit of each configuration. It was found that the optimum configuration for crashworthy components was the octet-lattice structure, based on the maximum value of specific energy absorption. Therefore, in this research, the twisted form of lattice structure is used for the battery protection component. The two-layered and single octet-lattice structure designs can be seen in Figure 3.

The optimization was carried out by varying three variables, i.e., the lattice angle, *θ*, relative density, $\bar{\rho }$*,* and material strength, $\sigma_{y}$. Relative density is defined by the volume ratio between the apparent truss–lattice structure and the continuous solid base material in which the cell edge is made.

The general equation of the relative density function is given as
(1)$$\bar{\rho }=k{\left(\frac{R}{t}\right)}^{2}-c\;{\left(\frac{R}{t}\right)}^{3}$$
where
(2)$$k=\pi *\frac{\sec\left(\frac{\theta }{360}\pi \right)+2\sqrt{2}*tan\left(\frac{\theta }{360}\pi \right)\;}{{tan}^{2}\left(\frac{\theta }{360}\pi \right)}$$
(3)$$c=\frac{\bar{c}}{{tan}^{2}\left(\frac{\theta }{360}\pi \right)}$$
(4)$$\bar{c}=\frac{\pi *R^{2}*Overall_{length}-Volume3D\_CAD}{R^{3}}$$

During optimization, the structure’s height was kept constant at 7.5 mm for each layer of the lattice. Therefore, the relative density is only determined by the angle and radius of the structure.
(5)$$\bar{\rho }=f\;\left(\theta ,R\right)$$

The samples provided for the neural network training process consisted of 30 samples for topology optimization and 18 samples for material optimization. This generates total permutations of 6 for lattice angle, 5 for relative density, and 3 for yield strength variations.

### 2.3. Modeling Process

#### 2.3.1. Loading and Boundary Conditions

Loading in the simulation was applied by an impactor that is perpendicular to the sandwich panel. Based on Halimah et al. study [11], the speed of 45 m/s was used as the initial velocity of the impactor. This number was assumed to be the most severe speed that could possibly occur from the debris hitting the underbody. Xia et al. [7] explained the kinematics of object impact. Furthermore, given the speed limit laws in the USA, the maximum allowable speed of a car on the highway is 85 mph [19]. The effect of wheeling, which adds centripetal speed, was also incorporated to represent a real-world accident. This was also done to prevent fatalities due to an extremely high-speed impact such as in Mexico in 2013 [20] and in Miami, USA in 2018 [21].

The impactor was defined to move in z-axis direction only, and no impactor rotation is assumed. The floor and sandwich panel edges were fixed in translation to represent real conditions in the car. Details of applied boundary conditions are listed in Table 1.

#### 2.3.2. Material Model

Material optimization of lattice structure was carried out by variations of the yield strength, which also affects the plastic region, as shown in Figure 4. The material properties were based on polymer HPPA12 (HP Polyamide 12) [22] with a basic yield strength of 22.8 MPa. Other larger hypothetical yield strengths of 45.6 MPa and 68.4 MPa were used for the material optimization study. The material properties of all components are shown in Table 2.

Shell elements were used to model all components except the jellyroll, impactor, and lattice components. Lattice structures were modeled using solid tetra elements. The automatic contact algorithm was used to define all connections between the components.

## 3. Lattice Optimization

### 3.1. Problem Description

The results of the finite-element analyses were then used for optimization by an artificial neural network (ANN) combined with genetic algorithms (GA) to find the value of weight and bias by minimizing the error. The genetic algorithm is expected to form the results of calculations, with weight and bias becoming more accurate since it often looks for global optimization. According to Gowda et al. [23], a neural network combined with the genetic algorithm has more accurate data results than using the back-propagation neural network method.

A neural network was constructed to derive the output variables of the energy absorption, SEA, and deformation, *δ*, as functions of input variables of relative density, $\bar{\rho }$, lattice angle, *θ*, and yield strength, *σ_y_*. Optimization processes, as depicted in Figure 5, were conducted to find the optimum points of the space design variables with the objective of maximizing SEA and minimizing *δ*, defined as in Equation (6).
(6)$$\left\{\begin{array}{c} Max\;\left(SEA\right)\;Min\left(\delta \right)\\ 35\leq \theta \leq 90\\ 0.25\leq \bar{\rho }\leq 0.80\\ 20\leq \sigma_{y}\leq 70 \end{array}\right.$$

### 3.2. Artificial Neural Network

The Multilayer Feed-Forward or Multilayer Perceptron (MLP) Neural Network Algorithm was used in this study. MLP has three parts—input, hidden, and output layers—each defined by neurons. The input-layer neurons represent the lattice angle, *θ*, relative density, $\bar{\rho }$, and yield strength, *σ_y_*. Two different MLPs were generated to determine SEA and deformation as a single output. The hidden-layer neurons are the parts trained by sample data to generate the function of output from the input data. A typical example of (MLP) ANNs with one input layer, two hidden layers, and one output layer is shown in Figure 6. The illustration of MLP network architecture of 3-7-2-1 is shown in Figure 7.

The MLP algorithm proceeds in three steps, i.e., dataset preparation, neural network construction, and combining with genetic algorithm. In the dataset preparation, a five-level full factorial design was used for selecting sample points within the design space of *θ*, $\bar{\rho }$, and *σ_y_*. Accordingly, 48 sample points were chosen for topology and material optimization.

The neural network was then constructed through the process of data training and testing on various hidden neurons and activation functions to obtain the optimum form. The hyperbolic tangent sigmoid activation function, Equation (7), was used for the hidden neuron.
(7)$$f\left(n\right)=tansig\left(n\right)=\frac{2}{1+e^{-2n}}-1$$

There is no particular rule about the necessary number of hidden layers and neurons. As shown by Stathakis [24], ANN configurations generally have two hidden layers. Therefore, in this study, two hidden layers were used for optimization. The number of hidden neurons was determined by splitting the dataset. In this study, 70% of the total data were taken randomly as training data, and the rest (30%) were used for testing purposes. In this case, the number of neurons was varied. An artificial neural network was trained using the training data. The testing data were used to calculate NMSE for every ANN. The data were trained five times and average results were taken to minimize the effect of sampling.

The NMSE is an estimator of the overall deviations between predicted and measured values. It is defined as:(8)$$\begin{array}{c} NMSE=\frac{1}{N}\;\underset{i}{\sum }\frac{{\left(Pi-Mi\right)}^{2}}{\bar{P}\;\bar{M}}\\ \bar{P}=\frac{1}{N}\underset{i}{\sum }Pi\\ \bar{M}=\frac{1}{N}\underset{i}{\sum }Mi \end{array}$$
where *N*, Pi, and Mi respectively refer to number of dataset, numerical result, and the output value predicted by neural network.

The number of neuron nodes in each hidden layer was determined as follows:(9)$$Max.\;first\;hidden\;neuron=\sqrt{\left(m+n\right)*N}+2\sqrt{N/\left(m+2\right)}$$
(10)$$Max.\;second\;hidden\;neuron=m\sqrt{N/\left(m+2\right)}$$
to give the maximum number of nodes that can be used to avoid overfitting. With the input value *n* = 3, the output value *m* = 1, and the number of total samples *N* = 48, we can have a maximum of 21 and 4 of first and second hidden-layer neurons, respectively. After being optimized, the numbers were reduced into 14 and 2 of first and second hidden-layer neurons, respectively.

Combined with the constructed ANN system, the GA method with 89 variables was then used to calculate the values of weight, w, and bias, b of the resulting ANN. Tuning parameters were used as shown in Table 3. With the global scope of optimization searching, the GA is expected to give more accurate results. The objective function of this genetic algorithm is to minimize the error using NMSE. The NMSE is an estimator of the overall deviations between predicted and measured values.

### 3.3. Multi-Objective Optimization

Multi-objective optimization was established by adjusting the design variables *θ*, $\bar{\rho }$, and *σ_y_* to find the optimum objective functions of the non-linear equations of crashworthiness indicators, SEA and *δ*, in terms of the design variables *θ*, $\bar{\rho }$, and *σ_y_*. A technique using a multi-objective genetic algorithm, termed NSGA-II (non-dominated sorting genetic algorithm II), was employed in this study to optimize optimal points of the structures. The searching of an optimum point was on the envelope, set by Pareto frontier points. The points were generated using tuning parameters as listed in Table 3.

The TOPSIS method was then implemented on all the Pareto frontier points to find optimum points. It is worth noting that in the TOPSIS method, we can apply weight or an important factor to each crashworthiness criteria. The four scenarios were considered as follows:*W_SEA_* = 0.4 and *W_δ_* = 0.6*W_SEA_* = 0.3 and *W_δ_* = 0.7*W_SEA_* = 0.2 and *W_δ_* = 0.8*W_SEA_* = 0.1 and *W_δ_* = 0.9
where *W_SEA_* and *W_δ_* present the weightings of SEA and *δ*, respectively. Deformation is weighted with higher value since it is considered more critical than SEA when designing energy absorbers for electric vehicle batteries.

## 4. Results and Discussions

### 4.1. Numerical Model Analysis

There were 48 samples for training data used in the optimization process. However, only one sample case will be discussed in detail, i.e., the lattice structure with *θ* = 42.45°, $\bar{\rho }$ = 0.462, and *σ_y_* = 22.8 Mpa. The deformation result obtained from numerical simulation is shown in Figure 8. The worst-case location for the ground impact point is selected at the center of the plate, which exhibits maximum deformation. Therefore, the middle battery located at this position is the most critical, with the greatest risk of being damaged or punctured. From Zhu’s research, the maximum shortening deformation that may occur is 3 mm. Beyond this deformation threshold, the battery will experience a short circuit, which leads to thermal runaway failure [8].

Figure 9 shows a sectional plane of the damage mode on the battery sub-system model. The blue transparent color shows the horizontal truss in the lattice. Although the deformed lattice showed general compressed behavior, the stretching deformation that occurs in the horizontal truss was the indication of stretching dominant deformation mode. The result is promising, since there is no discontinuity in the energy absorption process. The discontinuity often causes inefficiency in the progressive deformation of lattice structures, as shown by Nasrullah et al. [17].

In this ground-impact loading condition, represented by a stone impingement, the energy absorbed by the lattice structure must be maximized so that it can protect the battery from the ground impact. However, the energy absorbed by the lattice structure is not evenly distributed over the impact plane. This is due to the geometrical penetration of the impactor in the lattice structure. The structural sandwich construction, with a core lattice configuration, acts as an elastic foundation that distributes the impact load to improve the battery protection mechanism.

The energy absorbed by the sandwich structure can be seen in Figure 10. The discontinuity on the graph indicates a failure in the sandwich component. The first failure at 0.25 ms was the floor component failure. At about 1 ms, failure occurred in the lattice structure. The battery-shortening in Figure 10 shows reduced displacement behavior due to the elastic rebounding deformation of the jellyroll. This makes displacement of the jellyroll relatively lower due to a relieved pressure from the failure of the lattice structure.

### 4.2. Model Validation

The quality of the finite-element results can be verified by looking into the global energy curves, as shown in Figure 11. The finite-element result is deemed acceptable with the following energy characteristics: the total energy remains constant, the hourglass (error) energy is very small, and the sliding energy is not negative. Based on the energy characteristics shown in Figure 11, the dynamic finite-element computation of the battery simulation exhibits a very high-quality result.

### 4.3. Simulation Results

The results of the numerical simulations for all variations in 48 samples are shown in Table 4. It can be seen that there is no simple trend in the correlation between the topology and material parameters with the output of energy absorption and deformation of the sandwich panel. This is an example of cases where ANN and GA methodology can provide the solution for an optimization.

Simulations for the material with a yield strength of 22.8 MPa were performed for all variations of lattice angle and relative density to give the necessary behavior for the training process of ANN. The results, as shown in Table 4, provide sufficient data such that additional simulations at other yield strengths, i.e., 45.6 MPa and 68.4 MPa, were performed at selected lattice angles and relative density configurations.

### 4.4. Predictive Modeling Using ANN-GA

Genetic algorithms were applied to support the ANN to obtain the lowest possible NMSE (normalized mean square error) value of weight and bias. The error decreases as the number of generations increases. Multi-objective optimization in this study aims to minimize *δ* and maximize SEA.

### 4.5. Crushing Behavior

Parametric studies were carried out to investigate the effects of dimensional parameters *θ* and $\bar{\rho }$ on the crash performance. Figure 12a shows variations of the SEA against the geometrical parameters angle, *θ*, and $\bar{\rho }$ for the structures. The red dot indicates the optimum point obtained from the previous calculation. Figure 12b displays variations of the *δ* against *θ* and $\bar{\rho }$ for lattice structures.

The effects of the yield strength on the SEA and deformation, to represent material aspects, are shown in Figure 13a,b, respectively. The point marks show the data obtained from the FEM simulations, while the lines show the results of ANN prediction. As shown, the behaviors show that the effect of the material on the geometrical structure does not have a regular pattern.

Meanwhile, the effects of relative density and lattice angle parameters on internal energy absorption of the lattice structure component are more regular, as shown in Figure 14a,b. These results show the energy capacity of the lattice structure increases with the increase of relative density, lattice angle, and yield strength. The increasing of material volume, which follows the increase of relative density, will increase the energy absorption capacity. The lattice angle, as shown in Figure 15, is also directly proportional to the diameter of lattice trusses. With a larger diameter of trusses, the lattice structure will become stronger to resist the impact loading. Finally, the higher material yield strength implies higher plastic energy capability to absorb the impact loading energy.

### 4.6. Optimum Point

The combined ANN-GA technique was adapted to the TOPSIS method to derive the optimal lattice structure configuration for the abovementioned four scenarios. Table 5 presents the results of TOPSIS evaluation such that all weighting variation have the same results for optimum configuration point. Rounding up the values from Table 5, the optimum configuration of the lattice structure is a twisted-octet lattice structure with an angle of 66°, a density ratio of 0.8, and yield strength of 41 MPa.

The optimum configuration obtained by ANN-GA was then simulated by FEM to evaluate the accuracy of the predicted models (see Table 6). The structural geometry of the optimized octet-lattice battery protector can be seen in Figure 16. The damage to the optimized lattice battery protector is shown in Figure 17. The error between the predicted and FEM results was below 5%. This indicates that the optimization by ANN-GA was able to predict the crushing behavior of the structures. The error value of this study is greater than a previous study [18] with a similar approach. This may be because the geometry in this study is more complicated than in [18]. It may take a lot of datasets for the ANN error to be minimized. However, the present study has advantages over [18] because it has a reason to determine the number of neurons in the ANN.

With the objective of maximizing SEA and minimizing deformation, the effectiveness of the optimization method can be measured by the ratio of energy absorption to deformation, SEA: *δ*. As shown in Figure 18, the optimum configuration has the highest ratio compared to all sample designs—about 16% higher than the next-highest one. Compared to other studies, the value of SEA: *δ* obtained in this study is 3.8 times better than Halimah et al. [11], 3.5 times better than Nirmala et al. [12], and 2.1 times better than Daniel et al. [13].

## 5. Conclusions

In this study, the crashworthiness performance of the newly designed twisted-octet lattice structure configuration was evaluated for battery protection applications. A non-linear finite-element simulation was used to obtain the crashworthiness indicators against ground-impact loadings of a sandwich protection panel, namely energy absorption capability (SEA) and battery crash resistance (*δ*). The topology of the lattice structures (*θ* and $\bar{\rho }$) and the strength of material (*σ_y_*) were optimized using artificial neural networks and genetic algorithms.

The lattice structure energy capacity increases with an increase in relative density, lattice angle, and yield strength. The increasing material volume that follows increasing relative density will increase energy absorption capacity. The effect of increasing lattice angle is increasing diameter of lattice trusses, which makes the lattice structure better at resisting impact loading. Finally, the higher material yield strength implies higher plastic energy capability to absorb the impact of loading energy.

The varying neuron processes in the development of the ANN system have resulted in two hidden layers with 14 and 2 neurons for the first and second layers, respectively. The applied artificial neural network worked very well, indicated by a small value of normalized mean square error (NMSE). The genetic algorithm was successfully applied to create weight and bias with the smallest possible error limit.

Pareto frontier was used to define the envelope of optimum points in a multi-objective case, which is then evaluated by the TOPSIS method using four weighting scenarios of energy absorption (*W_SEA_*) and battery-shortening (*W_δ_*). Considering that battery-shortening is more critical than SEA for battery safety, the values of *W_δ_* were defined to be larger than the *W_SEA_*. Results of the TOPSIS method showed that the optimum lattice structure configuration was the one defined by a lattice angle of 66°, relative density of 0.8, and yield strength of 41 MPa. This configuration produces 3642 kJ/kg of SEA and 2.73 mm of battery deformation to give the highest SEA: *δ* ratio of all the sample data. Validation by FEM simulation of the performance of optimum lattice configuration, as predicted by ANN-GA, showed a small discrepancy below 5%.

The result showed the optimum structure to protect a Li-ion battery within the safety threshold of 3 mm battery-shortening. Therefore, the proposed lattice structure configuration can be implemented for electric vehicle applications to protect the battery from ground impact. For future work, we need to test the lattice structure performance experimentally to validate the effectiveness of the simulation.

## Figures and Tables

**Figure 1 materials-14-07618-f001:**
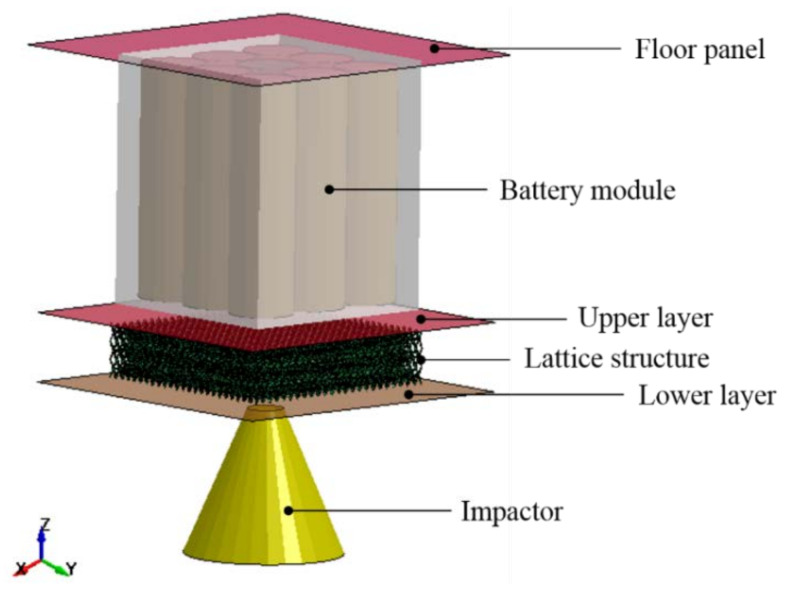
Numerical simulation model.

**Figure 2 materials-14-07618-f002:**
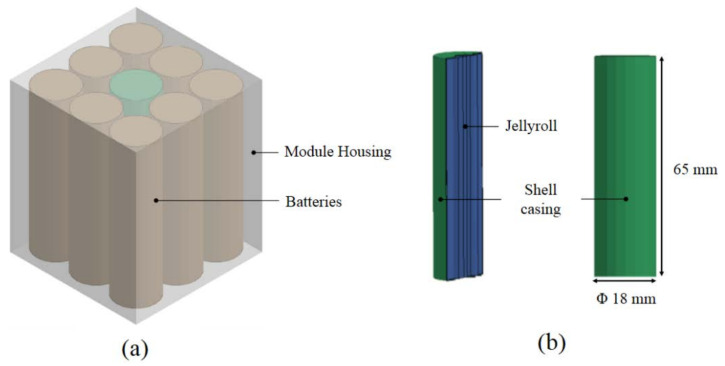
Battery (**a**) module model and (**b**) 18650 model dimensions.

**Figure 3 materials-14-07618-f003:**
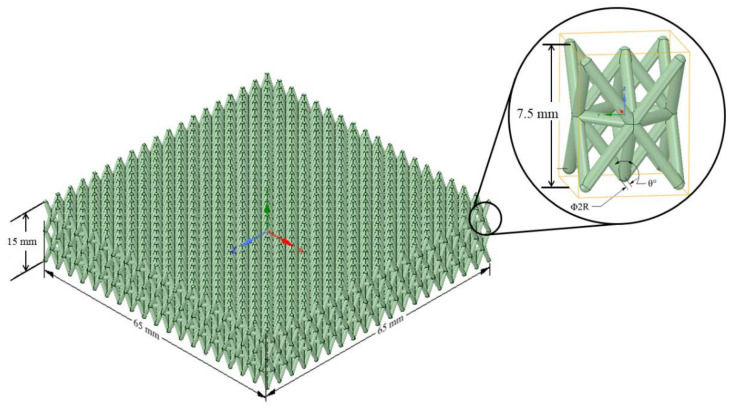
Geometry of cellular twisted-octet lattice structures.

**Figure 4 materials-14-07618-f004:**
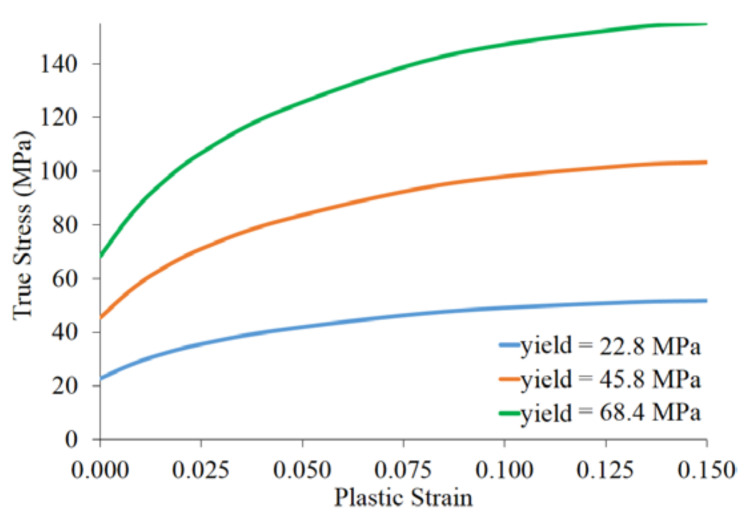
Material properties of lattice core.

**Figure 5 materials-14-07618-f005:**
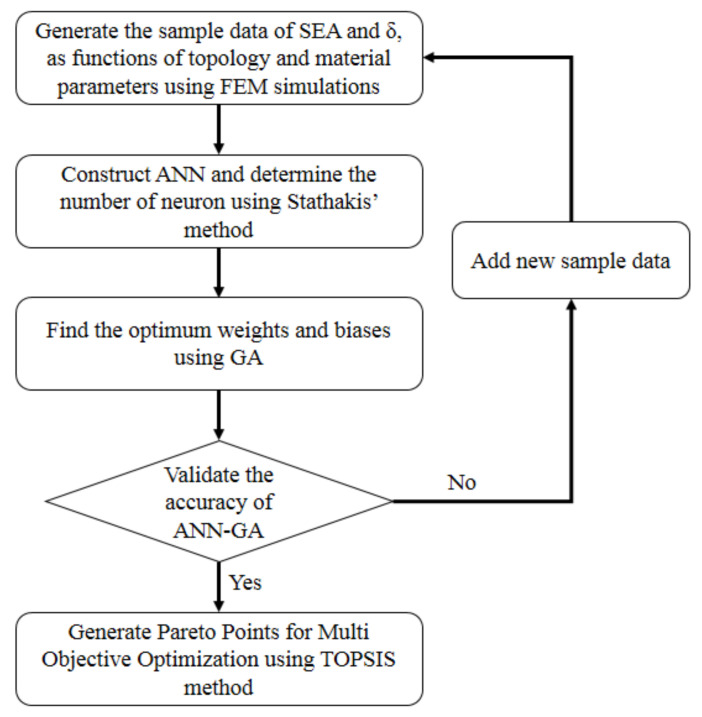
Flowchart of the optimization process.

**Figure 6 materials-14-07618-f006:**
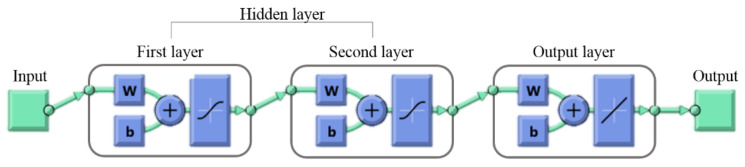
ANN block diagram of MATLAB.

**Figure 7 materials-14-07618-f007:**
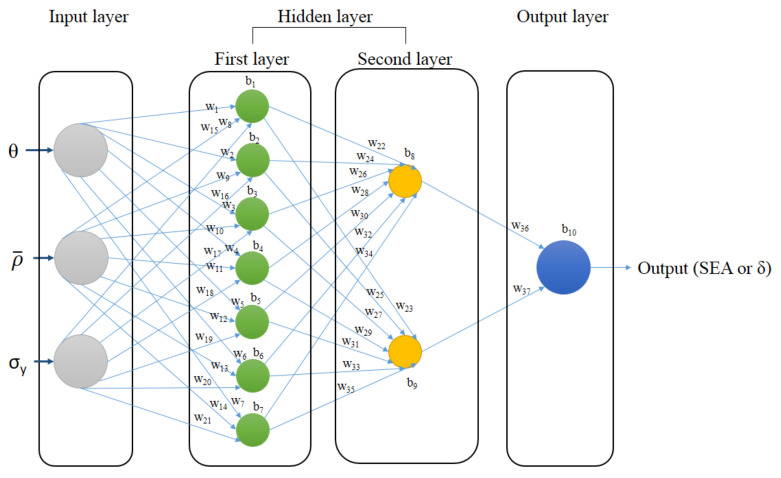
MLP network architecture illustration.

**Figure 8 materials-14-07618-f008:**
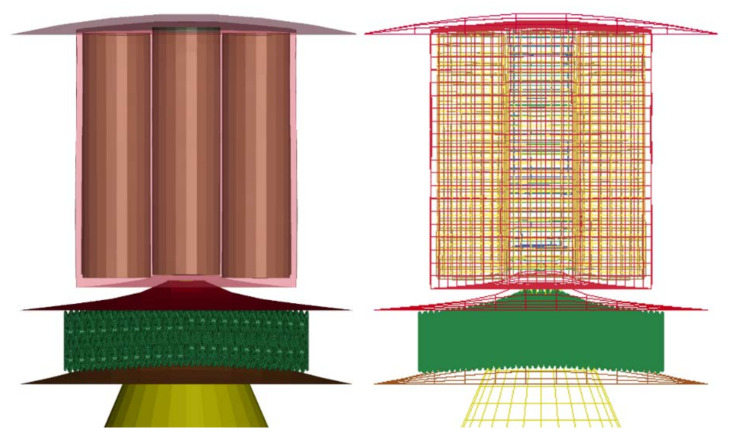
Deformation of battery.

**Figure 9 materials-14-07618-f009:**
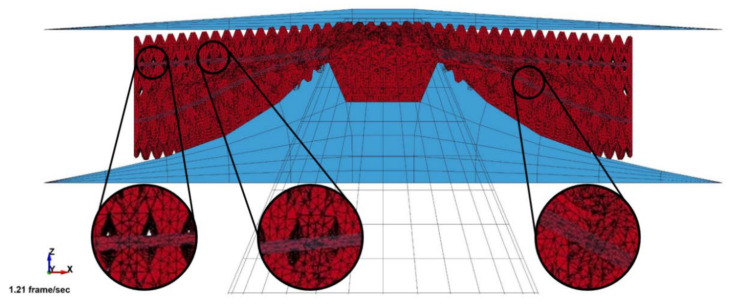
Section view of damage on the sandwich structure.

**Figure 10 materials-14-07618-f010:**
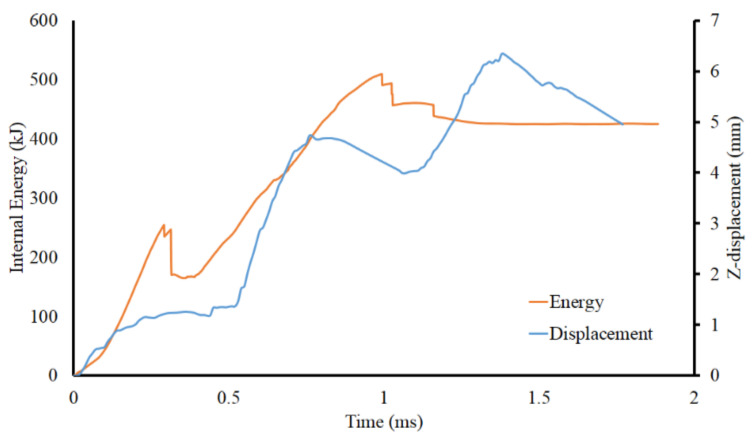
Energy absorption of baseline sandwich panel and displacement on battery jellyroll.

**Figure 11 materials-14-07618-f011:**
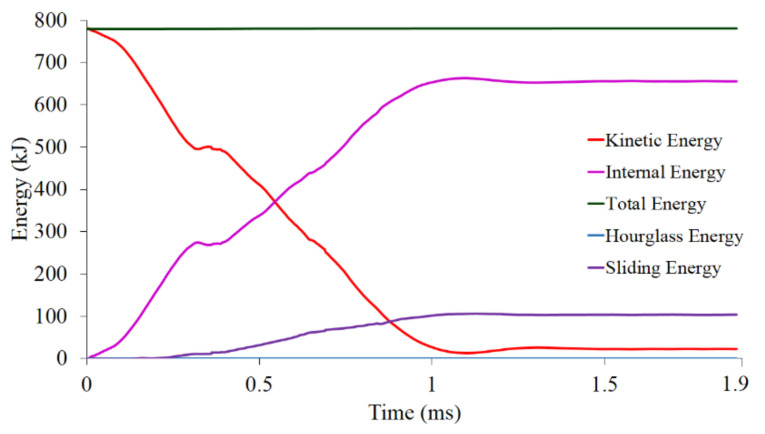
Battery energy validation curves.

**Figure 12 materials-14-07618-f012:**
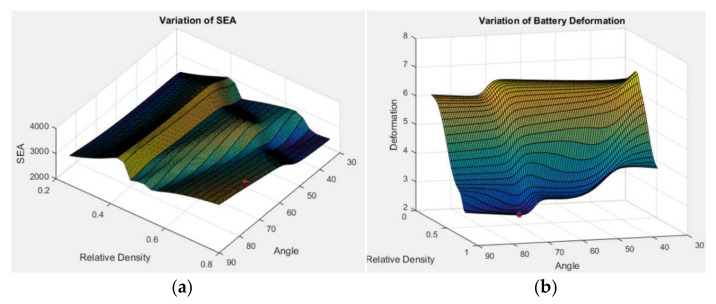
The variation response of: (**a**) SEA; (**b**) deformation, *δ* as functions of *θ* and $\bar{\rho }$.

**Figure 13 materials-14-07618-f013:**
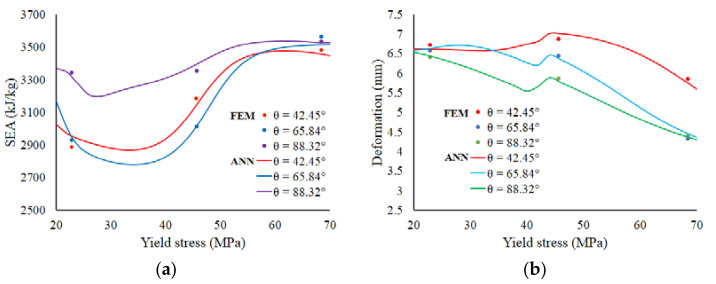
The crushing behavior in terms of: (**a**) SEA; (**b**) deformation, *δ*, as functions of yield strength at constant $\bar{\rho }$ = 0.273.

**Figure 14 materials-14-07618-f014:**
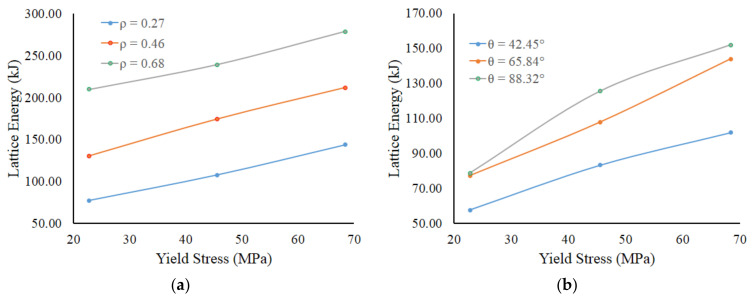
The energy absorption capacity of lattice structure as function of yield strength with parameters: (**a**) relative density at *θ* = 65.84°; (**b**) lattice angle at $\bar{\rho }$ = 0.27.

**Figure 15 materials-14-07618-f015:**
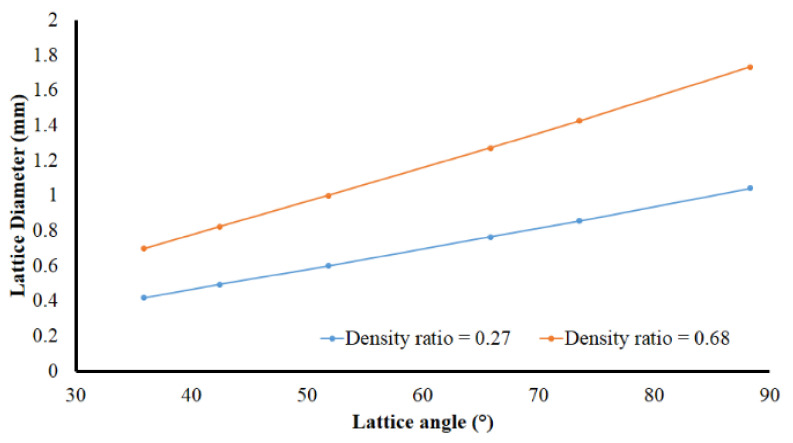
Diameter of lattice trusses as function of lattice angle at constant densities.

**Figure 16 materials-14-07618-f016:**
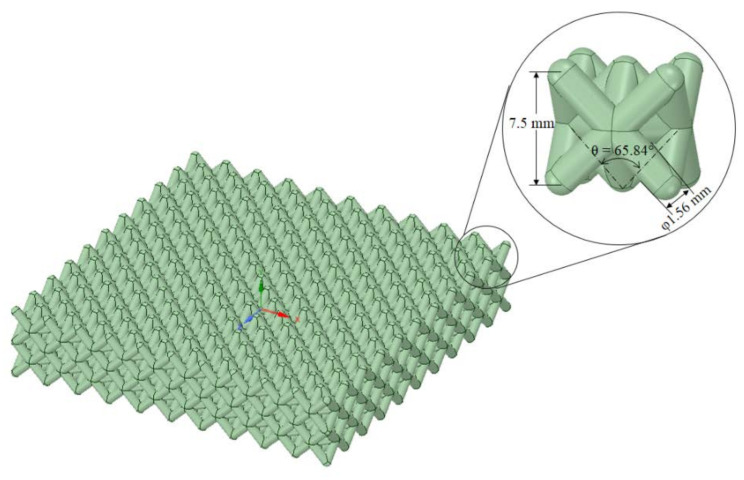
Geometry of optimized twisted-octet lattice structures.

**Figure 17 materials-14-07618-f017:**
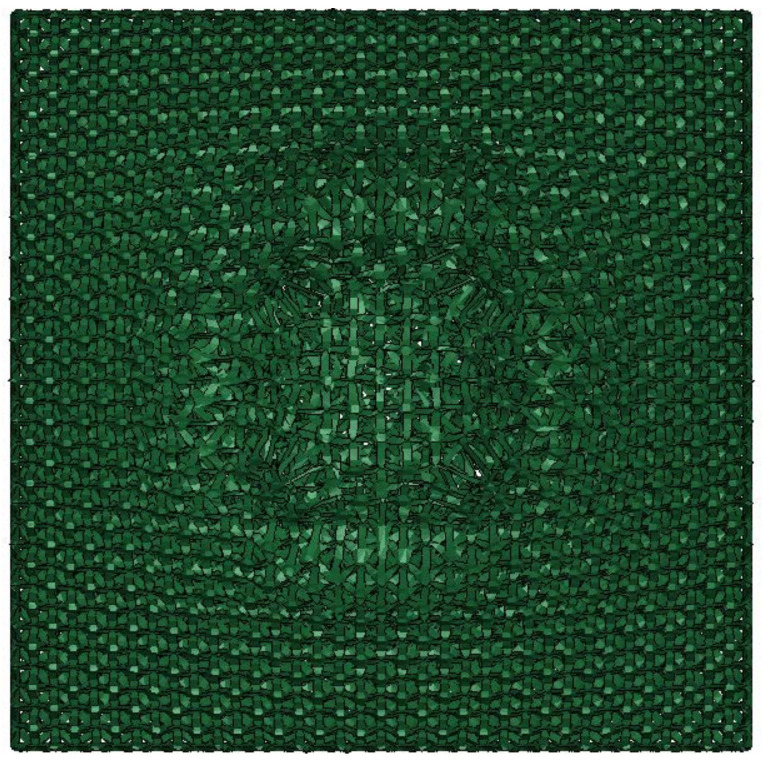
The damage of the optimized lattice structures after impact.

**Figure 18 materials-14-07618-f018:**
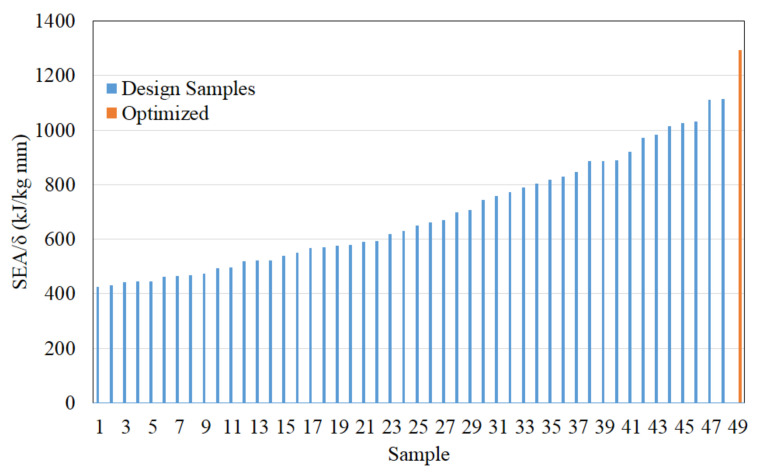
The ratio of energy absorption to the deformation to measure optimization effectiveness.

**Table 1 materials-14-07618-t001:** Boundary conditions.

Component	Location	Type of Boundary Condition
Floor panel	Edge	No translation (111000)
Sandwich panel	Edge	No translation (111000)
Impactor	All geometry	Only z-axis translation (110111)
Lattice	Edge	No x- and y-axis translation (110000)

**Table 2 materials-14-07618-t002:** Material properties.

Component	Material	Properties
Floor panel, sandwich panel	Aluminum alloy (AA-2024-T351)	*ρ* = 2.78 g/cm^3^ *σ_y_* = 324 MPa *δ_break_* = 20% *E* = 73.1 GPa 𝜈 = 0.33
Shell casing	Steel	*ρ* = 7.8 g/cm^3^ *σ_y_* = 450 MPa *E* = 200 GPa 𝜈 = 0.3
Module housing	Polypropylene	*ρ* = 0.905 g/cm^3^ *σ_y_* = 25 MPa *E* = 1.2 GPa 𝜈 = 0.42
Impactor	Rigid	*ρ* = 30 g/cm^3^
Jellyroll	Crushable foam	*ρ* = 2.721 g/cm^3^ *E* = 0.5 GPa
Lattice	HPPA12	*ρ* = 0.919 g/cm^3^ *σ_y_* = 22.8 MPa *E* = 1.34 GPa 𝜈 = 0.33

**Table 3 materials-14-07618-t003:** Tuning parameter of GA.

Tuning Parameter	Value
Population size	100
Maximum number of generations	10,000
Probability of cross over	0.8
Probability of mutation	0.01
Fitness limit	e^−5^
Initial population range	[−1;1]

**Table 4 materials-14-07618-t004:** Design sample points and numerical results for the structure with different lattice configurations.

*Θ* (°)	$\bar{\rho }$	*σ_y_* = 22.8 MPa		*σ_y_* = 45.6 MPa		*σ_y_* = 68.4 MPa	
		SEA (kJ/kg)	*δ* (mm)	SEA (kJ/kg)	*δ* (mm)	SEA (kJ/kg)	*δ* (mm)
35.87	0.27	2853	6.41				
35.87	0.36	3192	6.86				
35.87	0.46	2938	6.92				
35.87	0.57	3121	6.30				
35.87	0.68	3419	4.60				
42.45	0.27	2890	6.73	3187	6.88	3484	5.86
42.45	0.36	3514	6.11				
42.45	0.46	3426	6.60	2972	4.81	3463	3.91
42.45	0.57	3296	5.08				
42.45	0.68	3035	4.53	3202	3.12	3620	4.42
51.80	0.27	3148	6.65				
51.80	0.36	3466	7.01				
51.80	0.46	3301	6.13				
51.80	0.57	3472	4.97				
51.80	0.68	3000	3.54				
65.84	0.27	2934	6.58	3018	6.45	2565	4.34
65.84	0.36	3437	6.58				
65.84	0.46	3480	6.14	3398	3.46	3399	3.69
65.84	0.57	3507	4.97				
65.84	0.68	3386	3.48	3260	3.67	3349	4.42
73.52	0.27	2936	6.63				
73.52	0.36	3470	6.32				
73.52	0.46	3490	5.55				
73.52	0.57	3490	4.52				
73.52	0.68	3426	3.08				
88.32	0.27	3348	6.42	3359	5.88	3538	4.41
88.32	0.36	3477	6.02				
88.32	0.46	3551	5.37	3501	3.45	4155	3.73
88.32	0.57	3298	4.18				
88.32	0.68	3691	3.58	3113	3.75	3598	4.06

**Table 5 materials-14-07618-t005:** Optimum points evaluated by TOPSIS.

$W_{\delta }$	$W_{SEA}$	*θ* (°)	$\bar{\rho }$	$\sigma_{y}\;(\mathrm{MPa})$	SEA (kJ/kg)	*δ* (mm)
0.9	0.1	65.84	0.79	40.52	3642.80	2.73
0.8	0.2	65.84	0.79	40.52	3642.80	2.73
0.7	0.3	65.84	0.79	40.52	3642.80	2.73
0.6	0.4	65.84	0.79	40.52	3642.80	2.73

**Table 6 materials-14-07618-t006:** Comparison of predicted and FEA results for the optimum configuration.

	*θ* (°)	$\bar{\rho }$	$\sigma_{y}\;(\mathrm{MPa})$	SEA (kJ/kg)	*δ* (mm)
ANN-GA	65.84	0.79	40.52	3642.80	2.73
FEA	65.84	0.79	40.52	3712.41	2.87
Error (%)				1.87	4.87

## Data Availability

The data presented in this study are available on request from the corresponding author.

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
