# Peer review of "Structural Lattice Topology and Material Optimization for Battery Protection in Electric Vehicles Subjected to Ground Impact Using Artificial Neural Networks and Genetic Algorithms"

_materials, 2021, doi:10.3390/ma14247618_

Round 1

Reviewer 1 Report

The paper entitled “Structural Lattice Topology and Material Optimization for Battery Protection in Electric Vehicle Subjected to Ground Impact Using Artificial Neural Network and Genetic Algorithm” is well written and logically structured. It is interesting to see the applications of topology optimization, ANN, and GA to the electric vehicle. However, there are still some confusing statements, and some parts are not explained in detail. The Reviewer recommends the publication of this paper after major revision. The comments are listed as below:

  1. In section 2.2, please discuss more about lattice structure design to highlight what kind of design method you are using and why you use it. The authors can mention several lattice structures and then do a comprehensive comparison. In the end, the authors can give a conclusion on using the mentioned lattice method. The contents regarding lattice design is 2.2 is too little given lattice design is an important part in this paper.
  2. Why the authors use GA instead of PSO or other similar methods? Please provide some reasons in the paper.
  3. Figure 5 is the screenshot. Please use the original image.
  4. Fig. 9 a and b can be merged into one figure double y axis.
  5. Degrees in Fig 13 and Fig 14 b should not be “0”. Please change them.
  6. The resolution of Fig. 15 should be further improved.
  7. Please provide some optimized lattice structure, for example contour plots. That is, geometry of the new design should be given.
  8. It is obvious that this paper is simulation-based, however, for a simulation or design based paper, it is better to provide some experiments to validate the effectiveness of your simulation results. At least, add some discussions about your future validation work.

Author Response

Thank you for reviewing the paper. Please see the attachment.

Reviewer 2 Report

The authors have presented structural lattice topology and material optimization for battery protection using artificial neural network and genetic algorithm in this paper. the authors have claimed that the appropriate lattice structure is selected through topology and material optimization using artificial neural network (ANN), genetic algorithms (GA), and multi-objective optimization with TOPSIS methods. The paper is well written; however, some issues should be clarified in this paper. I recommend this paper for publication after some major modifications as follows:

  • The main issue about this paper is that how the ANN is used for the optimization? As the authors have claimed, the sample data were generated by using finite element. Also, the GA algorithm, which is an optimization algorithm is used for finding the weights of the MLP neural network. However, the MLP ANN could not be used as the optimization algorithm, because the ANN cannot perform extrapolation, while it can be applied for interpolation; so, it is suitable for modeling not optimization. Also, it cannot be concluded from the Flowchart in Fig. 5 that how did ANN perform the optimization in the proposed approach. The author should explain this issue for more clarification for readers.

  • In line 281 it is mentioned that the "The number of hidden neurons was determined by cross-validation method". The authors should explain how are the number of neurons determined using cross-validation method. In addition, “Cross-validation is a resampling procedure used to evaluate machine learning models on a limited data sample”.
  • The conventional MLP uses back propagation method to train the neural network. Why the GA is used instead of back propagation algorithm? In most of cases the MLP with backpropagation has more precision, compared with the MLP-GA network. Also using GA in MLP makes network very time consuming to run.
  • Provide equations for the errors used for the network. Also, for better clarification, illustrate the network architecture in a new figure and indicate the input and output parameters on it.
  • Some figures have low quality and some of them are not suitable for publishing. For example, Figs. 11 and 12. Also, in Fig. 5, correct the red underlined word.
  • Provide a comparison between the results of the proposed approach and similar works to show the contribution and novelty of the proposed paper.

Author Response

Please see the attachment. Please see the attachment.

Round 2

Reviewer 1 Report

This paper has been further improved.The Reviewer recommend the publication after the minor revision. My comments are:

In Figure 10, please put the legends inside the figure.

After Figure 16, please add the contour results of optimized design after impact. This was suggested before.

Author Response

(The authors gave the same response as above.)

Reviewer 2 Report

The comments are addressed in the revised manuscript and the quality of the new manuscript is now improved. However, some defects still exist, which should be corrected.

  • In line 295 ref [24] is mistakenly cited as [23].
  • The authors have claimed: " The comparison between the results of the proposed approach has been provided in Section 4.6. by comparing the error value". But there is not any comparison with the other similar works in this sections. I think the comparison of the proposed approach with related works should be improved in this section.
  • Also in section 4.6 line 452 it is written that: " The error value of this study is greater than the previous study [18] with similar approach". What did you mean by greater value of error in this work? If the errors are grater in this work with similar approach, so what is the advantage of your work compared to [18]? This statement should be clarified and more explained.   

Author Response

(The authors gave the same response as above.)
